# A Deep Learning Framework for Cardiac MR Under-Sampled Image Reconstruction with a Hybrid Spatial and *k*-Space Loss Function

**DOI:** 10.3390/diagnostics13061120

**Published:** 2023-03-15

**Authors:** Walid Al-Haidri, Igor Matveev, Mugahed A. Al-antari, Mikhail Zubkov

**Affiliations:** 1School of Physics and Engineering, ITMO University, Saint Petersburg 191002, Russiam.zubkov@metalab.ifmo.ru (M.Z.); 2Department of Artificial Intelligence, College of Software & Convergence Technology, Daeyang AI Center, Sejong University, Seoul 05006, Republic of Korea

**Keywords:** magnetic resonance imaging (MRI), medical image reconstruction, deep learning, conditional generative adversarial networks (CGANs), parallel imaging, hybrid spatial and *k*-space loss function

## Abstract

Magnetic resonance imaging (MRI) is an efficient, non-invasive diagnostic imaging tool for a variety of disorders. In modern MRI systems, the scanning procedure is time-consuming, which leads to problems with patient comfort and causes motion artifacts. Accelerated or parallel MRI has the potential to minimize patient stress as well as reduce scanning time and medical costs. In this paper, a new deep learning MR image reconstruction framework is proposed to provide more accurate reconstructed MR images when under-sampled or aliased images are generated. The proposed reconstruction model is designed based on the conditional generative adversarial networks (CGANs) where the generator network is designed in a form of an encoder–decoder U-Net network. A hybrid spatial and *k*-space loss function is also proposed to improve the reconstructed image quality by minimizing the L1-distance considering both spatial and frequency domains simultaneously. The proposed reconstruction framework is directly compared when CGAN and U-Net are adopted and used individually based on the proposed hybrid loss function against the conventional L1-norm. Finally, the proposed reconstruction framework with the extended loss function is evaluated and compared against the traditional SENSE reconstruction technique using the evaluation metrics of structural similarity (SSIM) and peak signal to noise ratio (PSNR). To fine-tune and evaluate the proposed methodology, the public Multi-Coil *k*-Space OCMR dataset for cardiovascular MR imaging is used. The proposed framework achieves a better image reconstruction quality compared to SENSE in terms of PSNR by 6.84 and 9.57 when U-Net and CGAN are used, respectively. Similarly, it demonstrates SSIM of the reconstructed MR images comparable to the one provided by the SENSE algorithm when U-Net and CGAN are used. Comparing cases where the proposed hybrid loss function is used against the cases with the simple L1-norm, the reconstruction performance can be noticed to improve by 6.84 and 9.57 for U-Net and CGAN, respectively. To conclude this, the proposed framework using CGAN provides the best reconstruction performance compared with U-Net or the conventional SENSE reconstruction techniques. The proposed framework seems to be useful for the practical reconstruction of cardiac images since it can provide better image quality in terms of SSIM and PSNR.

## 1. Introduction

Magnetic resonance imaging (MRI) is a safe and non-invasive diagnostic technique that does not use ionizing radiation to produce the images [1]. MRI is known to provide clear and detailed images of soft-tissue structures. These advantages make MRI a very effective diagnostic tool for a variety of disorders. However, the scanning procedure in modern MRI systems is very time-consuming. This leads to problems with patient comfort and causes motion artifacts [2,3]. The MRI system encodes the spatial information via the use of the magnetic field gradient in the scanned area, which results in assigning specific resonant frequency and local intensity to a specific anatomical area of the patient during the acquisition stage of MR imaging. The inverse Fourier transform is then applied to the encoded signals, which represent the phase-time space or the *k*-space, to produce an image [4]. Accelerated or parallel MRI has the potential to reduce medical costs and minimize patient stress. It is performed by reducing the number of gradient encoding steps during image acquisition and uses the information about the spatial locations and sensitivity profiles of the employed receiving coils to reconstruct the desired image. This allows for a shorter patient stay in the MRI scanner, which in turn decreases motion artefacts risk. However, such a reduction in the encoding step number results in the aliasing effect due to the violation of the Nyquist sampling requirements. In order to restore the desired image, parallel imaging techniques employ extra post-processing steps [5]. GRAPPA and SENSE are the dominant approaches to accelerated MRI, allowing for the effects of incomplete encoding in the phase-time domain and image domain to be mitigated via using the sensitivity maps of the receiving coils [6,7]. The SENSE technique first uses the inverse Fourier transform over the under-sampled *k*-space, followed by the restoration of the MR image, while GRAPPA estimates the fully sampled phase-time data and then uses Fourier transform to obtain an image [8]. Nevertheless, some information is lost in the under-sampling process, causing the signal to noise ratio (SNR) to drop and the reconstruction of artifacts.

The motivation of this work is to improve the quality of MR under-sampled image reconstruction by applying deep learning techniques. We hypothesize this would allow us to overcome the drawbacks of GRAPPA and SENSE, as the precision of MR image reconstruction has been shown to significantly increase when the latter employs recent deep learning developments. A crucial part of implementing a deep learning model to a particular task is finding a suitable loss function, sensitive to the difference between the generated and the target image. In this study, to enhance the quality of MR image reconstruction, we proposed a new hybrid spatial and *k*-space loss function which combines two loss functions in the spatial and frequency domains. The results of MR image reconstruction show that the developed hybrid loss could increase the precision of investigated models compared to the conventional L1-norm loss function. The developed models were trained and evaluated using the public Multi-Coil *k*-Space Dataset for Cardiovascular Magnetic Resonance Imaging, called OCMR [9]. Assessing the quality of cardiac MRI reconstruction was performed using the evaluation metrics of the structural similarity (SSIM) [10,11] and the peak signal to noise ratio (PSNR) [12]. Our deep learning approach allowed us to avoid the well-known parallel imaging effect of the SNR reduction, as the square root of the acceleration factor helped to mitigate this problem. The suggested algorithms and network architectures can therefore be used in applications where both fast acquisition and a high SNR is critical (with cardiac imaging being one of these areas) as a substitution for the classic parallel imaging algorithms, which do not conserve the SNR. Moreover, we investigated two state-of-the-art U-Net and CGAN networks and compared the reconstruction results using the proposed hybrid loss function against the conventional ones. The latter include L1-norm and GAN loss functions. The performance was evaluated on the OCMR [9] dataset using SSIM and PSNR metrics. The major objectives and contributions of this work are summarized as follows:
A new AI-based accurate parallel imaging reconstruction framework was proposed for better CMR image reconstruction.A new hybrid spatial and *k*-space loss function was proposed, which improves the SNR by taking into account the difference between the target (ground-truth, GT) and the reconstructed images in both spatial and frequency domains.Comprehensive reconstruction experimental studies were conducted with the aim to select the best AI model for the proposed framework. The model search additionally included the direct implementation of the conventional loss functions as well as their comparison with the proposed hybrid loss function.

## 2. Related Work

In this section, we summarize the recent deep-learning-based reconstruction methods for various MR imaging techniques. Based on the classical architecture of the deep network U-Net, Chang Min Hyun et al. [13] developed an algorithm for MR image reconstruction. Cardiac and brain images, obtained using the classical full *k*-space sampling technique, were used. The key point of their work was to combine the U-Net model with the following *k*-space data correction. The U-Net takes a folded image obtained from under-sampled zero-padded *k*-space data as input and recovers the zero-padded part of the *k*-space data. Then, the unpadded parts are replaced by the original *k*-space data to preserve the original measured data. Finally, inverse Fourier transform is performed to obtain the final reconstructed MR images. The developed algorithm showed decent results, with an average 0.90 SSIM. However, the complexity of the algorithm led to significant memory constraints when generating high-resolution images, which severely limited the output image resolution. Ghodrati et al. [14] investigated two CNN architectures: a simplified version of U-Net and the residual network (ResNet) for cardiac MR image reconstruction. The effect of four loss functions was investigated: pixel-wise L1 and L2, patch-wise structural dissimilarity (DSSIM), and feature-wise perceptual loss. According to the 57th quartile of SSIM score (0.88), U-Net–DSSIM (U-Net with DSSIM loss) performed significantly better than ResNet with different combinations of loss functions. However, U-Net has ten times the number of trainable parameters compared to ResNet, which results in increases in its computational complexity and computational time. 

A GAN-based algorithm was developed for knee joint image restoration from the reduced *k*-space without a reference fully sampled image [15]. The authors changed the concept of the classical GAN structure, where the generator network output is compared to the fully sampled data, by letting the generator serve as a seed for imitating the imaging process via subjecting the generator output to coil sensitivity map multiplication, FFT, and a randomized under-sampling mask. As a result, this produced sparsely sampled *k*-space data, which could be compared to the experimentally acquired sparse data. The proposed unsupervised GAN had superior PSNR, normalized root mean-square error (NRMSE), and SSIM compared to the common compressed sensing reconstruction. The unsupervised GAN only had 0.78% worse PSNR, 4.17% worse NRMSE, and equal SSIM compared to the supervised GAN. Reinforcement learning (RL) has also found an application in the field of medical image analysis, particularly in the reconstruction of brain and knee MR images [16]. The approach differs from the classical deep learning (DL) techniques in that MRI reconstruction is formulated as a Markov Decision Process—with discrete actions and continuous action parameters. An agent in such a process is a separate neural network that is assigned to each pixel of the MR image and processes it according to the reward received at each step of the algorithm training. The reward is formed as the difference between the values of the processed pixels at step *s* and (*s* − 1). The results on fastMRI data using a random 40% under-sampling mask were PSNR of 30.3 dB and SSIM of 88.0%.

The examples above, as well as this work, utilize the deep learning architecture network called U-Net which was used for biomedical image processing at the beginning of 2015 and has since shown the most powerful results. The U-Net structure is symmetrical and is divided into two main sections: the left half is called the encoder or contracting path and is made up of the basic convolutional processes, while the right section is known as the encoder or expansive path and is made up of transposed 2D convolutional layers [17]. Another candidate architecture considered in this work is the conditional generative adversarial network (CGAN), a modification of the conventional GAN. The GAN is one of the best neural network architectures for image processing and analysis, particularly for image synthesis and reconstruction [18]. Along with the complexity of some of the considered algorithms, a common drawback of the above methods is the low value of the signal to noise ratio. We assume that developing a custom hybrid loss function which calculates the difference between target and reconstructed images in both spatial and frequency domains will allow the model to overcome the drawback in recent works, i.e., the increase in the SNR, and lead to the high structural similarity of constructed images. 

## 3. Materials and Methods

### 3.1. The Proposed RecCGAN Framework: End-to-End Execution Scenario 

In this section, we describe the abstract view of the proposed AI-based framework for cardiac MR under-sampled image reconstruction using the hybrid spatial and frequency loss function. The proposed end-to-end workflow is presented in Figure 1 and is explained as follows:The fast MRI raw *k*-space data are collected and transformed into the spatial domain using the inverse fast Fourier transformer (IFFT).The MR images are resized into a fixed size of 256 × 256 pixels.After resizing, the FFT is applied to allow us to generate the under-sampled MR data in the frequency domain by removing each second column in the *k*-space domain (known as interleaved under-sampling).The IFFT is applied again to convert the under-sampled *k*-space data into the aliased MR images.All aliased images are normalized to fit all pixels within a fixed value range of [0, 255] to improve the AI learning process, and hence, the reconstruction performance. More detail about the data preparation can be found in Algorithm 1 (Section 3.2.2).The prepared aliased MR images are randomly split into 70% training, 10% validation, and 20% testing sets.To increase the number of training MR images, the augmentation strategy is applied to avoid any overfitting or bias, assist in better hyper-parameters’ optimization, and improve the reconstruction performance.For reconstruction purposes, two well-known deep learning architectures of U-Net and CGAN are adopted and used. The CGAN structure is adopted by using U-Net in an encoder–decoder fashion to build the generator network. However, we test and investigate the reconstruction performance of both U-Net and CGAN separately.The hybrid spatial and frequency loss function is proposed in order to improve the reconstructed image quality over conventional loss functions acting only in the spatial domain, such as L1-norm and GAN loss as a discriminator classification loss function.Finally, the proposed framework is evaluated using the individual U-Net and CGAN against the widely used conventional SENSE reconstruction algorithm. A direct, fair comparison is conducted using the same dataset and training environment settings.

### 3.2. Dataset

#### 3.2.1. Dataset Description

To build, train, and validate the proposed AI framework, the public Multi-Coil *k*-Space OCMR dataset for cardiovascular magnetic resonance imaging [9] was used. The dataset is available online at https://ocmr.info/download (accessed on 20 December 2022). The OCMR dataset consists of 53 fully sampled scans and 212 under-sampled scans. The fully sampled scans comprise 81 slices, while the under-sampled scans comprise 842 slices. These slices were collected from three different planes: 2-chamber, 4-chamber, and short-axis. To build and train the proposed AI reconstruction framework, the fully sampled scan data were used. The different available cine-frames were used as separate images, resulting in a total of 1383 multi-channel (from 15 to 35) full *k*-space data entries used for network training and testing.

#### 3.2.2. Dataset Preparation

Algorithm 1 shows the data preparation scenario for all 1383 multi-channel *k*-space data.
**Algorithm 1** Dataset preparation for parallel imaging simulation.**Start:**Input: Fully sampled *k*-space cardiac MRI data**Step 1**: load data*k*-space data = {‘kx’ ‘ky’ ‘kz’ ‘coil’ ‘phase’ ‘set’ ‘slice’ ‘rep’ ‘*avg*’} ← {read        *k*-space data in *.h5 format} ISMRMRD ← ISMRMRD; Python toolbox for MR image reconstruction [19]**Step 2**: Average the *k*-space data accumulations (kData) if ‘*avg*’ > 1*k-space data ← numpy.mean(kData, axis=* −1)**Step 3**: Apply IFFT to transform the *k*-space averaged data into the spatial domain*ImageSpaceData ← transform data from k-space into image space***Step 4**:Resize the MR image tensor for real and imaginary parts separately*Resized_image_tensor ← complex_mri_resize (ImageSpaceData, new_size)*Create a copy of Resized_image_tensor for under-sampling track*copy_resized_image_tensor ← numpy.copy(Resized_image_tensor)***Step 5**:Generate aliased MR imagesTransform the copy of resized image tensor back to the *k*-space*resized_kspace_tensor←transform_image_to_kspace(copy_resized_image_tensor])*2.Generate Cartesian binary sampling mask*Binary_mask ← numpy.zerose_like(resized_kspace_tensor)**Binary_mask[:..:*2*] =* 13.Under-sample the resized_kspace by removing every second column based on the designed binary sampling mask (Step 5: 2)*US_kspace ← resized_kspace_tensor * Binary_mask*4.Apply IFFT to get the under-sampled MR images*US_MR_image_tensor ← {transform_kspace_to_image(US_k-space)}*5.Merge the different channels via the sum-of-squares procedure*im_sos_full = numpy.sqrt(np.sum(np.abs(US_MR image_tensor) *** 2, 3*))*6.Remove singleton dimensions*Aliased_MR image_tensor ← numpy.squeeze(im_sos_full)***Step 6**:Generate fully sampled ground-truth (GT) MR imagesMerge the different channels (Step 4) via the sum-of-squares procedure*im_sos_full = numpy.sqrt(np.sum(np.abs(resized_image_tensor) *** 2*,* 3*))*Remove singleton dimensions*Fully sampled GT images ← numpy.squeeze(im_sos_full)***END**

Figure 2 shows the qualitative process of the MR data generation in terms of the reference or ground-truth (GT) images and aliased images. The IFFT was used twice to generate the GT and aliased MR images using the fully and under-sampled *k*-space data. To down-sample the fully sampled *k*-space data, the interleaved down-sampling strategy was used to generate the binary mask. Once the binary mask was generated with the same size as the *k*-space data, we multiplied it in the frequency domain with the original fully sampled data to generate the under-sampled *k*-space data. 

#### 3.2.3. MR Data Splitting and Augmentation 

Once the MR images were prepared in the spatial domain, the whole OCMR dataset was randomly split into 70% for training (887 *k*-spaces), 10% for validation (99 *k*-spaces), and 30% for testing (415 *k*-spaces). The 10% validation set was randomly picked from the training set. Table 1 shows the OCMR data distribution used to reach the goal of this study. For augmentation, we used random rotation, vertical and horizontal flipping, and cropping. 

### 3.3. Deep Learning Network Architecture and Training Details 

#### 3.3.1. U-Net with Hybrid Loss

We decided to investigate the U-Net model for MR image reconstruction due to its high efficiency in biomedical image processing. As mentioned above, the U-Net architecture comprises two parts [17]: the left part is the contracting path, in which a 3 × 3 convolution with zero-padding is applied for feature extraction from the input image. The rectified linear function is used as an activation function. Image down-sampling down to the bottleneck layer is conducted via a max-pooling layer with a 2 × 2 stride. The right section following the bottleneck layer is known as the expansive path and is made up of transposed 2D convolutional layers. To increase the reconstruction precision, the architecture suggests concatenating the up-sampled layer output on the expansive path with the corresponding feature tensor from the contracting path.

As the performance of the deep learning model depends not only on the network architecture, but also on the loss function, which plays an important role in minimizing the model error, a number of loss functions were assessed. The initial approach was to train the model with L1 loss function [13] and L2 regularization.
(1)$$J\left(\theta \right)=\frac{1}{N}\overset{N}{\underset{s=1}{\sum }}\|{ℋ}_{ℒ}^{\left(s\right)}-{\hat{ℋ}}_{ℒ}^{\left(s\right)}\|+\lambda \cdot \|\Theta^{2}\|,$$
where ${\hat{ℋ}}_{ℒ}^{\left(s\right)}$, ${ℋ}_{ℒ}^{\left(s\right)}$ are the model output and reference image, respectively, Θ is the tensor of trainable parameters, and λ is the regularization parameter. N denotes the batch size. To improve the model efficiency, we later extended the loss function by additionally taking into account the difference between target and reconstructed images in the frequency domain (i.e., in the *k*-space). The new loss term, which we call Fourier loss, is then provided by
(2)$${ℒ}_{1}^{ℱ}=\frac{1}{N}\overset{N}{\underset{s=1}{\sum }}\|ℱ\{{ℋ}_{ℒ}^{\left(s\right)}\}-ℱ\left\{{\hat{ℋ}}_{ℒ}^{\left(s\right)}\right\}\|,$$
where $ℱ\{{ℋ}_{ℒ}^{\left(s\right)}\}\;,\;ℱ\left\{{\hat{ℋ}}_{ℒ}^{\left(s\right)}\right\}$ are the Fourier transform of the reconstructed and the reference images, respectively. The suggested loss function was tested as a part of an extended loss function obtained by adding the Fourier loss (2) to Equation (1)
(3)$$J\left(\theta \right)=\frac{1}{N}\overset{N}{\underset{s=1}{\sum }}\|{ℋ}_{ℒ}^{\left(s\right)}-{\hat{ℋ}}_{ℒ}^{\left(s\right)}\|+\alpha \cdot \frac{1}{N}\overset{N}{\underset{s=1}{\sum }}\|ℱ\{{ℋ}_{ℒ}^{\left(s\right)}\}-ℱ\left\{{\hat{ℋ}}_{ℒ}^{\left(s\right)}\right\}\|+\lambda \cdot \|\Theta^{2}\|,$$
where 𝛼 = 0.1.

#### 3.3.2. CGAN with Hybrid Loss

The GAN is another neural network architecture well-suited for image processing and analysis, particularly for image synthesis and reconstruction. GANs consist of two competing networks: the first, the generator, is the network that transforms the random noise to generate ‘fake’ but realistic-looking images. The second network, called the discriminator, is a different network, trained to classify whether the images generated by the generator are real or ‘fake’. Here, we implemented the GAN with the U-Net architecture as a generator and a convolutional network as a discriminator. The discriminator network comprised six consecutive convolutional layers, four out six followed by batch normalization and all followed by an activation function (ReLU in the first five and sigmoid in the last layer). Another adopted modification to the classic GAN architecture fed the generator not with random noise, but with an MR image obtained from the reduced *k*-space [18]. This modification allowed such a model to be called image-conditional GAN (CGAN), as shown in Figure 3.

Training the generator comprises finding the minimum of the objective function. The objective specific to the CGAN is minimizing the loss function through the generator network and maximizing it through the discriminator.
(4)$$\underset{G}{\min}\underset{D}{\max}{ℒ}_{CGAN}\left(D,G\right)=E_{x,y}\left[\log D\left(x,y\right)\right]+E_{x,G\left(x\right)}\left[\log\left(1-D\left(x,G\left(x\right)\right)\right)\right]$$
where E[*arg*] is the mean of *arg*, and *D*(*x*, *y*) and *G*(*x*) are the discriminator and the generator functions, respectively. It is also recommended to add L1 or L2 distance between the reference and the generated image to the CGAN objective [18] to increase the method accuracy. We used the L1 distance here because it encourages less blurring [18]. Adding L1 distance changes the objective to
(5)$$Loss_{CGAN}=\underset{G}{\min}\underset{D}{\max}{ℒ}_{CGAN}\left(D,G\right)+\sigma \cdot E_{x,y}\left[\|y-G{\left(x)\right\|}_{1}\right],\;$$
where the coefficient 𝜎 is chosen empirically. As in the case of the standalone U-Net architecture, we have found it beneficial to use Fourier loss in the objective function of the CGAN: (6)$${ℒ}_{1}^{ℱ}=E_{ℱ\left(x\right),\;ℱ\left(G\left(x\right)\right)}\left[\|ℱ\left(y\right)-ℱ{\left(G\left(x\right))\right\|}_{1}\right].\;$$

Thus, identically to the bare U-Net case, two options for the loss function improvement were explored: the L1 norm and the Fourier loss as a part of the combined loss. The latter is given by
(7)$$Loss_{CGAN}^{*}=E_{x,y}\left[\log D\left(x,y\right)\right]+E_{x,G\left(x\right)}\left[\log\left(1-D\left(x,G\left(x\right)\right)\right)\right]+\sigma \cdot E_{x,y}\left[\|y-G{\left(x)\right\|}_{1}\right]\;+\alpha \cdot E_{ℱ\left(x\right),\;ℱ\left(G\left(x\right)\right)}\left[\|ℱ\left(y\right)-ℱ{\left(G\left(x\right))\right\|}_{1}\right].\;$$

The discriminator model is trained separately on fake data (pairs of images acquired from the reduced *k*-space and the corresponding CGAN-generated images) and real data (pairs of images acquired from the reduced *k*-space and the reference image). The two input images are concatenated together to create one 256 × 256 × 2 input to the first hidden convolutional layer. The discriminator training strategy is illustrated in Figure 4. 

In the training process, the discriminator model can be updated directly, whereas the generator model must be updated via the discriminator model. This can be achieved by creating a new composite model that connects the generator model’s output to the discriminator model’s input. The discriminator model can then predict whether a generated image is real or fake. To prevent a misleading update of the discriminator when employing a discriminator to update the generator, the discriminator weights are specified as not trainable [18,20]. Figure 5 illustrates the training strategy for the complete CGAN model with Fourier loss.

To evaluate the efficiency of the deep learning approach, we compared the deep-learning-based results with the SENSE reconstruction*,* which is one of the most widely used parallel imaging methods, offered by the majority of MR scanner vendors. As the coil sensitivity profiles are not the part of the OCMR dataset, they were estimated for the SENSE reconstruction from fully sampled MR data using the algorithm presented in [21]. 

### 3.4. Evaluation Strategy 

Two metrics were used to assess the quality of the cardiac MRI reconstruction in both the bare U-Net and CGAN cases. The first was the peak signal to noise ratio. The *PSNR* is a metric for assessing the degree of pixels’ distortion caused by compression and noise [12]. It is defined as
(8)$$PSNR=20\cdot \log_{10}\left(\frac{MAX_{I}}{\sqrt{MSE}}\right),$$
where $MAX_{I}$ is the maximum possible pixel value of the image, and MSE is the mean squared deviation.

Structure similarity (*SSIM*) is the second metric, which is more complex and more informative than the *PSNR*. It comprises an assessment of three characteristics of the investigated images: intensity, contrast, and structural difference [11].
(9)$$SSIM{\left(I,K\right)}_{local}=\frac{\left(2\mu_{I}\mu_{K}+c_{1}\right)\left(2\sigma_{I,K}+c_{2}\right)}{\left(\mu_{I}^{2}+\mu_{K}^{2}+c_{1}\right)\left(\sigma_{I}^{2}+\sigma_{K}^{2}+c_{2}\right)}$$
$$SSIM\left(I,K\right)=\frac{1}{M}\overset{M}{\underset{i=1}{\sum }}SSIM{\left(I,K\right)}_{local}$$
where SSIM(I,K)∈[0,1] is the structural similarity between the target image *I* and the generated image *K*. μ and σ are the average and standard deviation, respectively. c is the constant.

The SSIM value ranges from 0 to 1. If SSIM is 1, then the images are identical. As a rule, SSIM is not used for the entire image at once, but it is used locally with a sliding window and then averaged. The local calculation of SSIM metrics allows the variability in statistical characteristics and spatial heterogeneity of the image structure to be taken into account. Here, the kernel size was chosen to be (8 × 8) according to [10].

### 3.5. Experimental Setup

End-to-end training was used for the proposed AI model. In this study, we employed a learning rate of 0.0002 with Adam optimizer. We trained all AI models using 100 epochs with Random Normal weight initialization (stddev = 0.02) and a batch size of 8. The input and output image size was fixed to the dimensions of 256 × 256. On the encoder side, the activation function of LeakyReLU (alpha = 0.2) was used, while ReLU was used for the decoder side. 

### 3.6. Execution Development Environment 

A computer with the following specifications was used to carry out the experiments: AMD Ryzen 7 5800X 8-Core Processor 3.80 GHz 32 GB RAM with RTX 3060 (8 GB) GRAPHICS CARD. Python 3.10 running on Windows 10 along with the Keras and TensorFlow backend libraries were utilized to conduct the experiments that were analyzed in this study.

## 4. Experimental Results

The considered models were evaluated on the test MR data subset using the PSNR and the SSIM evaluation metrics. The latter employed the image reconstructed from the under-sampled *k*-space with deep learning models as I in (9) and the reference images obtained from the fully sampled data as K in (9). Figure 6 and Table 2 display comparisons of SSIM and PSNR metrics of the reconstructed test MR images using: the U-Net model with the L1-loss (*U-Net_L1*), the U-Net model with the combination of the L1- and Fourier loss (*U-Net_Hybrid_Loss*), the CGAN model with L1-loss (*GAN_L1*), and the CGAN model with the combination of the L1 and Fourier losses (*GAN_ Hybrid_Loss*). As an addition, the abovementioned models were compared to the SENSE parallel MR imaging algorithm reconstruction. 

### 4.1. Quality of the MR Image Reconstruction against Different AI Architectures

Our first goal was to study the impact of the deep network architecture type on the quality of the MR image reconstruction. For this purpose, the U-Net and the CGAN architectures were investigated. According to the median values of the evaluation metrics, the CGAN architecture exceeded the U-net by 2% in terms of the SSIM score, in which they reached the values of 0.89 and 0.87, respectively. The CGAN architecture also showed a better median PSNR score (33.91) when compared to the median PSNR of images reconstructed with the U-Net (32.77). Thus, the CGAN model showed overall better performance according to the SSIM and PSNR metrics.

### 4.2. Quality of the MR Image Reconstruction against the Proposed Hybrid Loss Function 

Our second goal was to explore the impact of the introduced modified loss function. The results in Table 2 and Figure 6 show the contribution of the proposed Fourier loss function, which was designed to take into account the difference between target and reconstructed images in the frequency domain. Adding Fourier loss to the U-Net and the CGAN model resulted in increases in the SSIM score by 2% and 3%, respectively, whereas the PSNR score increased by about 1.02 and 2.2 for the two network architectures. Thus, we can see that the proposed Fourier hybrid loss helped to enhance the image reconstruction quality. Making the model minimize the error in the *k*-space resulted in minimizing the reconstruction error.

## 5. Discussion 

### 5.1. Deep Learning Approach against Classical Algorithms of MR Image Reconstruction

The results above show the ability of the deep learning algorithms to reconstruct MRI images as an alternative to classical algorithms such as SENSE. Different deep network architectures (U-Net and CGAN) achieve better image reconstruction quality against SENSE in terms of the PSNR by 6.84 and 9.57, respectively. Similarly, the deep network architectures studied in this paper display comparable SSIM results of the reconstructed MR images. The advantage of the developed deep learning approach is that there is no need for coil sensitivity maps compared to the SENSE algorithms of the MRI image reconstruction. The higher value of PSNR, comparable SSIM metric, and the absence of the need for a coil sensitivity map, along with other possibilities, open up promising prospects for the development of deep learning approaches in MRI image reconstruction problems.

L1 distance in image space forces the model to enhance the structural characteristics of generated (reconstructed) images. Using Fourier (*k*-space) L1 loss, we encouraged the model to take important frequency components into account; thus, we achieved an increase in the quality of image reconstruction for both networks. Adding Fourier (*k*-space) loss to the U-Net and the CGAN model resulted in increasing the SSIM score by 2% and 3%, respectively, whereas the PSNR score increased by about 1.02 and 2.2 for the two network architectures. Thus, we can see that the proposed hybrid loss helps to enhance the image reconstruction quality.

### 5.2. Statistical Significance of the Results

After the PSNR and SSIM metrics were calculated using the test dataset for every architecture, the metrics distributions were tested for statistically significant differences. The choice of the statistical significance test depends critically on the type of the data distribution. As a rule, if the samples have a normal distribution, then a *t*-test is used. If the data distribution does not meet the requirements for normality, then other approaches are undertaken, the most commonly known being the Mann–Whitney U-test. It is therefore necessary to first conduct a test for data normality and then evaluate the statistical significance. Normality tests were carried out using a qualitative histogram evaluation as well as a Shapiro–Wilk normality test. The essence of normality verification is to put forward the null hypotheses that the data are distributed normally with the error probability of 0.05. Thus, H0 is “the data come from the normal distributions (accepted if *p* > 0.05)”; otherwise, H1 (rejected), meaning that data do not come from the normal distribution. The results of the Shapiro–Wilk normality test are presented in Table 3.

Since for all the architectures in Table 3, the *p*-values for SSIM and PSNR distributions were much less than the alpha (*p*-value 0.05), we rejected all the null hypotheses and concluded that none of the samples came from normal distributions. These results were also confirmed by the graphic evaluation. It can be seen in Figure 7 that the data were not distributed normally.

Due to the non-normality of the SSIM and PSNR distributions, we could not use a t-test to study statistical significance, and thus, a nonparametric Mann–Whitney U-test was used. We compared the SSIM and PSNR distributions provided by the final architecture (GAN_Hybrid_Loss) with the rest of the reconstruction strategies. The results of the Mann–Whitney U-test are presented in Table 4.

Using the Mann–Whitney U paired test with a significance level of a 0.05 *p*-value, the AI models were investigated. Assuming the null hypothesis, there was no significant performance difference between our proposed model (GAN_Hybrid_Loss) and others, whereas the alternative hypothesis intended to show if the proposed GAN_ Hybrid_Loss model provided metrics that were statistically different from the other approaches. According to the obtained *p*-values of the SSIM metric, shown in Table 4, we could infer that statistical differences existed between GAN_ Hybrid_Loss and other models (*p*-value much smaller than 0.05), except SENSE (*p*-value = 0.09 > 0.05). This shows the differences between the GAN_Hybrid_Loss model and other models (seen in Figure 6 and Table 2) were statistically significant differences in all cases except SENSE, which thus can be concluded to have had a comparable performance. On the other hand, the p-values of the PNSR distribution tests for all models were smaller than the 0.05 threshold, which confirms the improvement in the performance of our GAN_Hybrid_Loss model against other models, including the SENSE algorithm, to be statistically significant.

### 5.3. Comparison Results against the Recent Research Works 

During this study, we conducted a comparison of the proposed algorithm with the latest AI research works for MRI image reconstruction. Table 5 shows the used models, implemented loss functions, and some quantitative results of studied deep learning algorithms for MR image construction. The analysis of these approaches shows that regardless of how good the SSIM metrics are, the PSNR is still close to the mean PSNR of the classical SENSE algorithm. In [9], some *k*-space correction was used, but it was employed as a post-processing step, so the model itself did not learn this correction in the *k*-space. Unfortunately, the it did not provide any information about the PSNR, so it is difficult to evaluate the contribution of this post-processing procedure. However, our approach, thanks to the use of hybrid spatial and *k*-space loss, overcame the presented models in the PSNR metric (mean PSNR = 35.68). Thus, we can conclude that in MR image reconstruction, it is important to pay attention to the difference between reconstructed and target images not only in the special space, but also in the *k*-space. This will guarantee the achievement of the more accurate quality of the image reconstruction.

## 6. Conclusions

In this work, we developed a deep learning approach to reconstruct cardiac MR images from under-sampled *k*-space data. Two deep network architectures were considered: the U-Net and the CGAN. The results showed that the CGAN model outperformed the U-Net model by 2% in terms of the SSIM score. To enhance the model efficiency, we extended the loss function by additionally taking into account the difference between target and reconstructed images in the frequency domain. The proposed loss, referred to as the Fourier loss, was shown to increase the SSIM by another 2% for the U-Net model and by 3% for the CGAN. The PSNR score was also improved by employing the Fourier loss by 1 for the U-net model and by 2.2 for the CGAN model.

Because the GAN model with the combination of L1 and Fourier losses (*GAN_Hybrid_Loss*) yielded the best results among the other studied deep learning models, we also compared it with the reconstruction employing the SENSE algorithm. According to the SSIM metric, the results of *GAN_Hybrid_Loss* are comparable to the SENSE results. However, the PSNR of *GAN_Hybrid_Loss* was greater than that of the SENSE by 8.7 (36.11 and 27.40, respectively). The latter could have resulted from the known effect of SNR reduction in parallel imaging as the square root of the acceleration factor, while the deep learning algorithms do seem to help mitigate this problem. The suggested algorithms and network architectures can therefore be used in applications in which both fast acquisition and a high SNR is critical (cardiac imaging being one of these areas) as a substitution for the classic parallel imaging algorithms, which do not conserve the SNR. 

## Figures and Tables

**Figure 1 diagnostics-13-01120-f001:**
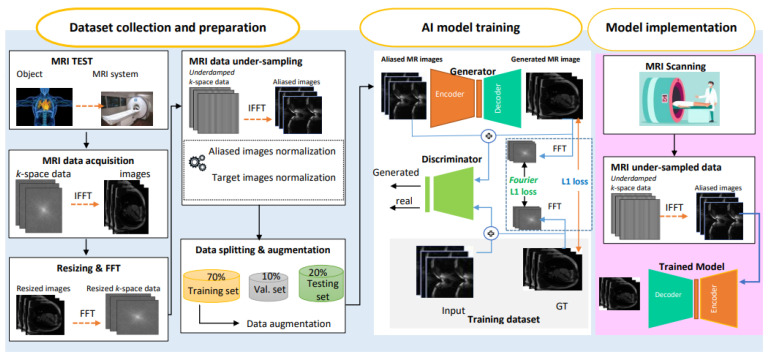
The proposed AI-based MRI reconstruction framework.

**Figure 2 diagnostics-13-01120-f002:**
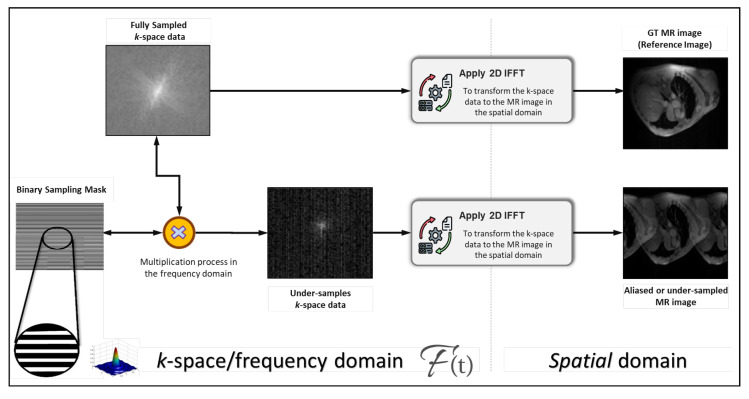
Generating the reference ground-truth (GT) and aliased MR images using the fully and under-sampled *k*-space data, respectively.

**Figure 3 diagnostics-13-01120-f003:**
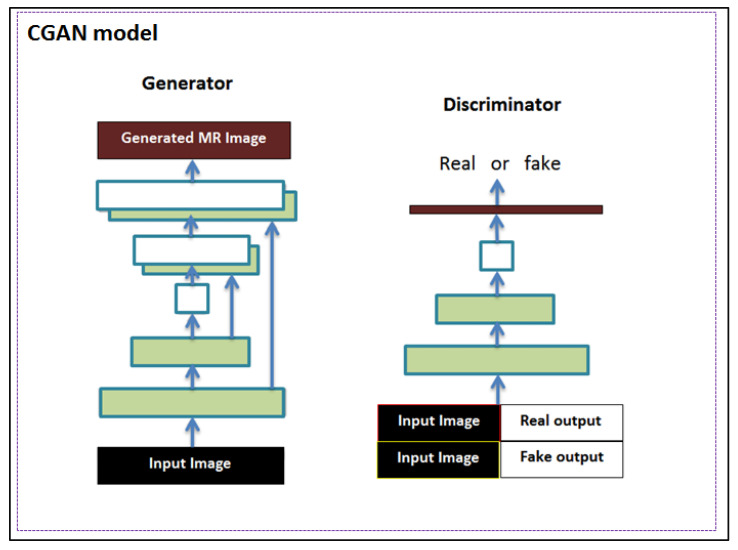
The CGAN model architecture. The generator employs the U-Net architecture, whereas the discriminator is a convolutional network.

**Figure 4 diagnostics-13-01120-f004:**
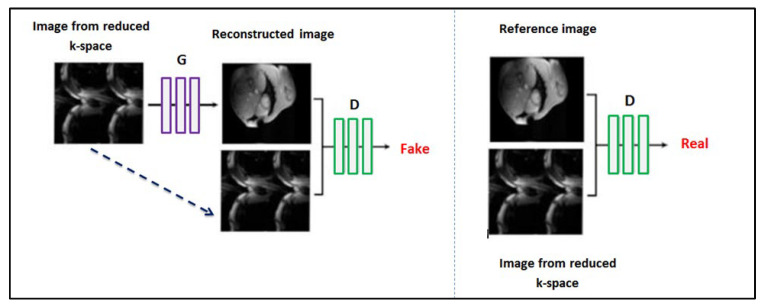
The CGAN model architecture. The generator employs the U-Net architecture, whereas the discriminator is a convolutional network.

**Figure 5 diagnostics-13-01120-f005:**
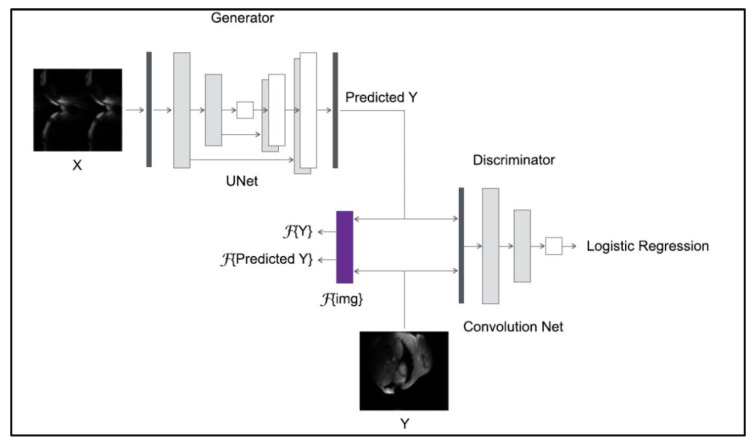
The training strategy for the CAGN model with Fourier loss.

**Figure 6 diagnostics-13-01120-f006:**
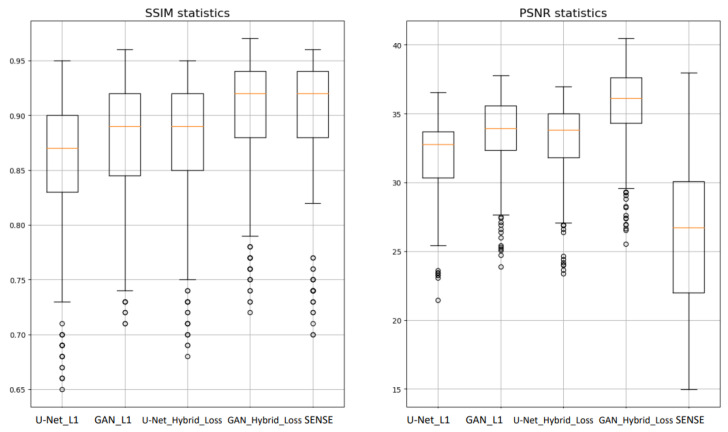
Comparison of the SSIM and PSNR metrics of the reconstructed test images for different algorithms: U-Net model with the L1-loss (*U-Net_L1*), U-Net model with the combination of the L1- and Fourier loss (*U-Net_L1_fft*), CGAN model with the L1-loss (*GAN_L1*), CGAN model with the combination of the L1 and Fourier losses (*GAN_L1_fft*), and the classic SENSE reconstruction.

**Figure 7 diagnostics-13-01120-f007:**
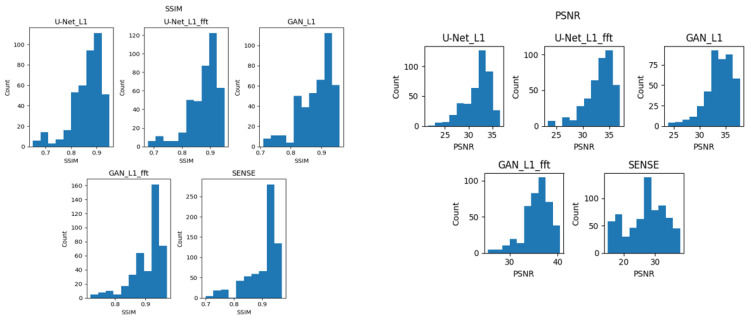
Graphical evaluation of SSIM and PSNR for studied models.

**Table 1 diagnostics-13-01120-t001:** MCOR data distribution.

	Training (70%)	Validation (10%)	Testing (30%)
Original Dataset	887	99	415
Augmented Dataset	3548		

**Table 2 diagnostics-13-01120-t002:** Reconstruction performance evaluation of the proposed methodology against SENSE over the test dataset.

AI Model	SSIM		PSNR		No. of Trainable Parameters (Million)	Training Time/Epoch (s)	Testing Time/Image (s)
	Mean ± SD	Median	Mean ± SD	Median			
U-Net_L1_Loss	0.857 ± 0.059	0.87	31.834 ± 2.66	32.77	54.41	1.23	0.1
GAN_L1_Loss	0.880 ± 0.054	0.89	33.678 ± 2.6	33.91	61.38	4	0.15
U-Net_Hybrid_Loss	0.876 ± 0.054	0.89	33.112 ± 2.56	33.79	54.41	1.23	0.1
GAN_ Hybrid_Loss	**0.903 ± 0.050**	**0.92**	**35.683 ± 2.77**	**36.11**	61.38	4	0.15
SENSE	0.902 ± 0.058	0.92	26.288 ± 5.48	26.70	-	-	0.29

**Table 3 diagnostics-13-01120-t003:** Shapiro–Wilk normality test for SSIM and PSNR of studied AI models.

AI Model	SSIM			PSNR		
	Statistics	df	*p*-Value	Statistics	df	*p*-Value
U-Net_L1_Loss	0.885	414	5.3 × 10^−17^	0.925	414	1.61 × 10^−13^
GAN_L1_Loss	0.921		7 × 10^−14^	0.943		1.66 × 10^−11^
U-Net_Hybrid_Loss	0.882		2.8 × 10^−17^	0.905		2.30 × 10^−15^
SENSE	0.819		5.75 × 10^−17^	0.954		1.11 × 10^−13^
GAN_Hybrid_Loss	0.868		3 × 10^−18^	0.941		9.46 × 10^−12^

**Table 4 diagnostics-13-01120-t004:** The results of Mann–Whitney U test for SSIM and PSNR of the studied AI models against the model with the highest SSIM and PSNR (GAN_Hybrid_Loss).

AI Model	SSIM	PSNR
	*p*-Value	*p*-Value
U-Net_L1_Loss	3.26 × 10^−38^	3.20 × 10^−74^
GAN_L1_Loss	3.83 × 10^−12^	2.02 × 10^−28^
U-Net_Hybrid_Loss	2.02 × 10^−18^	2.12 × 10^−43^
SENSE	**0.092**	2.40 × 10^−116^

**Table 5 diagnostics-13-01120-t005:** Comparison of the evaluation results against the latest AI research works for MRI image reconstruction.

Reference	Model	Loss Function	SSIM	PSNR
Hyun CM et al. (2017), [9]	U-net with *k*-space correction	L2-norm	0.903	-
Ghodrati V et al. (2019), [10]	Resnet-L1	L1-norm	0.81	26.39
Ghodrati V et al. (2019), [10]	Unet–Dssim	Structural dissimilarity	0.86	27.04
Cole, Elizabeth et al. (2020), [11]	Unsupervised GAN	GAN loss	0.88	29
The proposed, (U-Net_Hybrid_Loss)	U-Net	Hybrid Loss function **Hybridd Loss**	0.876 ± 0.03	33.11 ± 2.56
The proposed, (GAN_Hybrid_Loss)	**CGAN**		**0.903 ± 0.05**	**35.68 ± 2.77**

## Data Availability

The datasets used in this paper are publicly available at: https://ocmr.info/download/ (accessed on 20 December 2022).

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
