# Peer review of "A Deep Learning Framework for Cardiac MR Under-Sampled Image Reconstruction with a Hybrid Spatial and k-Space Loss Function"

_diagnostics, 2023, doi:10.3390/diagnostics13061120_

Round 1

Reviewer 1 Report

The authors submitted a research article in which they elucidated new deep-learning MR image reconstruction framework is proposed to provide more accurate reconstructed MR images when the under-sampled or Aliased images are produced. The authors OCMR dataset consists of 53 fully sampled scans and 212 under sampled scans. The fully 197 scans comprising 81 slices, while the under-sampled scans comprising 842 slices. They found that the proposed framework using CGAN provide the best reconstruction performance compared with U-Net or the conventional SENSE reconstruction techniques. Yet, they validated this technique. The manuscript has logic structure and composes of well-written subsections, which cover up all aspects of the initial hyppthesis. The figures and tables are clear and legible. The conclusive part is attractive for readers and contains informative issues. I would like to congratulate the authors on the study. However, I woud like to put forward some comments to discuss.

1. Fig. 6 (Comparison of the SSIM and PSNR metrics of the reconstructed test images for different algorithms). Statistics difference is missed. Please, this information

2. Tables 2 and 3: There is a lack of comparissons for the models. Please, modify this table accordingly

Reviewer 2 Report

The authors of the article want to propose a new deep-learning MR image reconstruction framework to provide more accurate reconstructed MR images when the under-sampled or Aliased images are produced. The proposed reconstruction model is designed based on the conditional generative adversarial networks (CGAN) where the generator network is designed in a form of encoder-decoder U-Net network. The hybrid spatial and k-space loss function is also proposed to improve the reconstructed image quality by minimizing the L1-distance considering both spatial and frequency domains simultaneously. The proposed reconstruction framework is directly compared when CGAN and U-Net are adopted and used individually based on the proposed hybrid loss function against the conventional L1-norm.

The proposed new reconstruction system is superior when compared to others, so this could partly revolutionize the way MR images are studied and reconstructed.

The article is well written, certainly full of technicalities, and suitable for a specialist audience, more engineering than medical.

The use of English is adequate but I would recommend a complete revision to streamline some of the lexical periods.

The bibliography is adequately written and thorough.

Round 2

Reviewer 1 Report

The authors submitted a revised version of the paper along with clear explanation of the ways by which they revised the manuscript. I have no serious concerns about the article in its revised version.

Reviewer 2 Report

It is agreed with the changes made.

Good work